# Organization of the Structural Protein Region of La Jolla Virus Isolated from the Invasive Pest Insect *Drosophila suzukii*

**DOI:** 10.3390/v13050740

**Published:** 2021-04-23

**Authors:** Tessa Carrau, Benjamin Lamp, Carina M. Reuscher, Andreas Vilcinskas, Kwang-Zin Lee

**Affiliations:** 1Department of Bioresources, Fraunhofer Institute for Molecular Biology and Applied Ecology, Ohlebergsweg 12, 35392 Giessen, Germany; tessa.carrau@ime.fraunhofer.de (T.C.); andreas.vilcinskas@ime.fraunhofer.de (A.V.); 2Institute of Virology, Justus Liebig University, Schubertstrasse 81, D-35392 Giessen, Germany; Benjamin.j.lamp@vetmed.uni-giessen.de (B.L.); Carina.m.reuscher@vetmed.uni-giessen.de (C.M.R.); 3Institute for Insect Biotechnology, Justus Liebig University, Heinrich Buff Ring 26-32, D-35392 Giessen, Germany

**Keywords:** *Drosophila suzukii*, La Jolla virus (LJV), *Iflavirus*, genome organization, capsid proteins

## Abstract

*Drosophila suzukii* (Ds) is an invasive pest insect that infests ripening fruit, causing severe economic losses. Control measures based on chemical pesticides are inefficient and undesirable, so biological alternatives have been considered, including native Ds viruses. We previously isolated a strain of La Jolla virus (LJV-Ds-OS20) from Ds in Germany as a candidate biopesticide. Here we characterized the new strain in detail, focusing on the processing of its capsid proteins. We tested LJV growth during Ds development to optimize virus production, and established a laboratory production system using adult flies. This system was suitable for the preparation of virions for detailed analysis. The LJV-Ds-OS20 isolate was cloned by limiting dilution and the complete nucleotide sequence was determined as a basis for protein analysis. The terminal segments of the virus genome were completed by RACE-PCR. LJV virions were also purified by CsCl gradient centrifugation and analyzed by SDS-PAGE and electron microscopy. The capsid proteins of purified LJV virions were resolved by two-dimensional SDS-PAGE for N-terminal sequencing and peptide mass fingerprinting. The N-terminal sequences of VP1 and VP2, together with MS data representing several capsid proteins, allowed us to develop a model for the organization of the LJV structural protein region. This may facilitate the development of new viral strains as biopesticides.

## 1. Introduction

*Drosophila suzukii* (Ds; Matsumura, 1931) was originally native to East and Southeast Asia but has spread beyond its native range due to global trade and transport. This invasive species, also known as the “spotted wing *Drosophila*”, is now an economically devastating pest of fruit crops in the Americas and Europe [1,2,3,4]. Whereas most of the 1500 known *Drosophila* species feed on rotting fruit, Ds females are attracted by ripening fruit [5] and deposit their eggs into intact fruits by punching holes in the skin with a specialized serrated ovipositor [6]. The high reproduction rate of ~400 eggs per female and the rapid life cycle of 8 days under optimal conditions enable rapid population growth, thus causing massive damage to fruit crops shortly before harvest [7,8]. The use of chemical pesticides against hatching Ds larvae is hampered by the natural protected habitat within the fruit, the timing of application shortly before ripening, and the rapid acquisition of resistance. Alternative strategies that are safe and effective are therefore needed urgently to control this invasive pest. Entomopathogenic viruses are often host specific and therefore have a limited impact on non-target organisms, making them attractive candidates for the development of environment-friendly control agents [9]. For example, *Cydia pomonella* granulosis virus (CpGV) is already used in apple orchards against the larvae of the codling moth (*Cydia pomonella*) [10].

We recently identified La Jolla virus (LJV, *Iflaviridae*) in Ds specimens captured in Germany [11], and others have detected the same virus species in Ds specimens from France, the UK and Japan [12]. Infection of adult Ds with the German LJV strains caused death within 3 days, making these strains promising candidate biopesticides. LJV was initially identified as a novel drosophilid *Iflavirus* based on metagenomic analysis [13]. The family *Iflaviridae* is composed of positive-sense single-stranded RNA viruses found exclusively in arthropods, and has been detected in a range of hemipteran, coleopteran and lepidopteran insects [14,15,16]. Members of the genus *Iflavirus* include economically important pathogens such as deformed wing virus (DWV), slow bee paralysis virus (SBPV) and sacbrood virus (SBV) [17]. The name is derived from infectious flacherie virus (IFV), the first *Iflavirus* described and the cause of infectious flacherie disease in silkworm (*Bombyx mori*) [18]. DWV is the etiological agent associated with the collapse of honeybee colonies and is transmitted by the ectoparasitic mite *Varroa destructor* [19]. The symptoms associated with DWV include malformations of honeybee pupae such as stunted wings, bloated abdomens, neurological disorders, and death [20].

The morphogenesis of iflaviruses has been studied in detail for the honeybee pathogens DWV and SBV, which feature non-enveloped virions ~30 nm in diameter [21,22,23]. The single-stranded RNA genome is enclosed within an icosahedral capsid comprising 60 protomers formed from the structural proteins VP1, VP2 and VP3. Unlike mammalian picornaviruses, *Iflavirus* capsids do not appear to contain VP4, but a comparable small VP4-like peptide of unknown function seem to be generated during polyprotein processing [24]. The *Iflavirus* genome features a single open reading frame (ORF) that is translated into a polyprotein. The latter is processed into its functional subunits by a viral 3C-like protease [13]. A covalently linked VPg protein, which is required for picornaviral RNA replication, is thought to be present at the 5′ end of the genome, and the 3′ end features a polyadenylated tail. *Iflavirus* infection generally follows the oral uptake of contaminated food, and the tropism of the viruses appears to include the intestine, gonads, fat body, muscle, brain and glandular epithelial cells.

Although DWV and SBV are well characterized [21,22,23], LJV has not been studied in detail and more information is needed for the development of effective biopesticide strains. The LJV genome is 10,250 nucleotides in length (excluding the 3′ polyadenylated tail). The genomic organization of the capsid protein region has not yet been described in detail with respect to the processing and boundaries of the individual structural proteins as functional components of the protomers. [14]. We therefore purified strain LJV-Ds-OS20 from Ds specimens captured in west-central Germany, allowing us to sequence the genome and investigate the processing of the capsid proteins, thus leading to a model of the structural protein region.

## 2. Materials and Methods

### 2.1. Insects and Viruses

The Ds line used for our experiments was derived from a laboratory colony established in southern Ontario during the summer of 2012 [25]. The insects were maintained in 2.5 cm diameter vials at 26 °C and 60% humidity (Figure 1A) with a 12-h photoperiod and were fed on soybean and cornmeal medium consisting of 10.8% (*w/v*) soybean and cornmeal mix, 0.8% (*w/v*) agar, 8% (*w/v*) malt, 2.2% (*w/v*) molasses, 1% (*w/v*) nipagin, and 0.625% propionic acid. The LJV-Ds-OS20 isolate discussed herein was isolated from wild Ds specimens captured in Germany as previously described [11].

### 2.2. Virus Infections

Adult female Ds were injected in the thorax (Figure 1B) with 46 nL of diluted virus master stock containing 10^10^ genome equivalents per milliliter (GE/mL). After incubation for 3–7 days at 26 °C, flies were anesthetized with CO_2_ and snap-frozen at −80 °C in vials containing 0.05% Triton X-100 in PBS.

### 2.3. Virus Preparation and Purification

To purify the virus, ~2000 adult female flies (2 g) infected as described above were homogenized using a bead beater homogenizer, FastPrep System, with 1.4 mm ceramic beads (MP Biomedicals, Eschwege, Germany). After two cycles (45 s at 6.5 m/s), the homogenate was cleared by centrifugation (3000× *g*, 15 min, 4 °C) to remove cell debris and residual chitin. The supernatant was then passed through a Rotilabo 0.45-µm PVDF syringe filter (Carl Roth, Karlsruhe, Germany). Virions were concentrated by ultracentrifugation on a 25% (*w/w*) sucrose cushion (~100,000× *g*, 4 h, 4 °C) using a Type 45 Ti rotor (Beckman-Coulter, IN, USA) at 30,000 rpm. The pellet was resuspended in 500 µL PBS by continuous agitation for 48 h at 4 °C. This virion concentrate was then cleared by centrifugation (15,000× *g*, 1 min, 4 °C) and layered onto a continuous 12 mL CsCl gradient. The initial linear gradient was generated with 0–100% of a 1.5 g/mL CsCl solution in PBS using an ÄKTA pure 25 FPLC system (GE Healthcare, München, Germany). After centrifugation (~200,000× *g*, 20 h, 4 °C) using a SW 41 Ti rotor (Beckman-Coulter, IN, USA) at 35,000 rpm, the gradients were illuminated with a basal light source and the bands were carefully removed and weighed on an ABS 80-4N analytical balance (Kern, Balingen, Germany). A second gradient also containing the virus was sampled from the top, and all fractions were measured using the analytical balance as a control.

### 2.4. Transmission Electron Microscopy

Transmission electron microscopy (TEM) was carried out as previously described [11]. Briefly, the purified particles were adsorbed onto carbon-coated grids and contrasted using 2% (*w/v*) uranyl acetate. Images were generated using a Biotwin CM120 device (Philips, Hamburg, Germany) connected to an SIS Olympus Keenview camera (Olympus, Tokyo, Japan).

### 2.5. SDS-PAGE, Edman Degradation and Mass Spectrometry

For 1D gel electrophoresis, protein samples were denatured in loading buffer, separated by sodium dodecylsulfate polyacrylamide gel electrophoresis (SDS-PAGE) on a 12.5% polyacrylamide gel, and stained with Coomassie Brilliant Blue R250 or transferred to a PVDF membrane. For 2D gel electrophoresis, protein samples were solubilized in denaturing buffer (6 M urea, 2 M thiourea, 4% CHAPS, 1% DTT and 2% Pharmalyte 3–10). IPG strips (pH 3–10) were rehydrated and the solubilized protein samples were fractionated by isoelectric focusing at 32.05 kVh. The IPG strips were then equilibrated for 10 min in 2 mL equilibration stock solution (6 M urea, 0.1 mM EDTA, 0.01% bromophenol blue, 50 mM Tris-HCl pH 6.8, 30% glycerol) and incubated for 15 min in 2 mL of the same solution containing 20 mg/mL SDS and 10 mg/mL DTT, then for 15 min in the same solution containing 20 mg/mL SDS and 48 mg/mL iodacetamide. Second dimension SDS-PAGE was carried out as above, followed by staining or transfer to a PVDF membrane.

N-terminal amino-terminal sequence analysis by automated Edman degradation on a gas-phase sequencer, Procise Model 491 (Applied Biosystems) in the Proteomics Facility, Justus Liebig University, by Dr. Guenther Lochnit. For matrix-assisted laser-desorption ionization time-of-flight mass spectrometry (MALDI-TOF-MS), protein spots were digested with trypsin after reduction and carbamidomethylation, then analyzed on an Ultraflex TOF/TOF mass spectrometer equipped with a nitrogen laser, LIFT-MS/MS, and Compass v1.4 software including FlexControl v3.4, FlexAnalysis v3.4 and BioTools v3.2. The instrument was operated in positive-ion reflectron mode with a matrix comprising 2,5-dihydroxybenzoic acid and methylendiphosphonic acid. Proteins were identified by peptide mass fingerprinting using MASCOT (Matrix Science, Boston, MA, USA) to screen the NCBInr and Uniprot databases as well as the translated ORF of the LJV genome. We applied a mass tolerance of 75 ppm, with cysteine carbamidomethylation accepted as a global modification and oxidation of methionine as a variable modification. A false positive rate of 5% was allowed.

### 2.6. Quantification of Viral Loads by qRT-PCR

Virus RNA was quantified by qRT-PCR on an ABI 7500 cycler (Applied Biosystems, Foster City, CA, USA) using the LJV-specific primers LVqRTfor570 (5′-AAG GCC TTG GAA ACC TTC ATC TC-3′) and LVqRTrev664 (5′-CAA ATA CTA CAC AGC CGA CCT CCA-3′), the FAM/TAMRA-labeled probe LVqRTprobe630 (5′-Fam-TCG TAT ATG ATG ATC ACA AGG TTG CTC ACA CC-TamRa-3′) and the Luna universal probe one-step RT-qPCR kit (New England Biolabs, Ipswich, MA, USA). A plasmid carrying the LJV-Ds-OS20 target cDNA sequence under the control of the SP6 promoter was linearized with MluI, purified by gel electrophoresis and quantified by spectrophotometry. A synthetic RNA fragment was transcribed from the cDNA template using the HiScribe SP6 RNA synthesis kit (New England Biolabs, Ipswich, MA, USA), purified using the peqGOLD total RNA kit (VWR PeqLab, Darmstadt, Germany) including the DNA removal step and quantified by spectrophotometry. A standard curve was prepared by including a 10-fold dilution series of the RNA control. Each reaction was initiated by heating to 55 °C for 10 min, then to 95 °C for 1 min, followed by 40 cycles of 95 °C for 10 s and 60 °C for 30 s (amplification and fluorescence detection step). The number of genome copies was calculated against the standard curve using ABI 7500 System SDS Software (Applied Biosystems).

### 2.7. Sequencing

Preparation of a cDNA library and Illumina sequencing was carried out by the Novogene Company (Beijing, China) on an Illumina NovaSeq 6000 platform including quality control and bioinformatics. A total of 4.3 Mio events (called bases) in 15,000 reads were generated and used to determine the LJV-Ds-OS20 consensus sequence as well as to exclude the presence of contaminating RNA viruses in the stocks. Contaminating viruses were also tested by pathogen-specific RT-PCR. Total RNA was extracted from virus suspensions using the QIAamp RNA Mini Kit (Qiagen, Hilden, Germany) and RT-PCR was carried out using the OneTaq One-Step RT-PCR Kit (New England Biolabs, Ipswich, MA, USA) or the One Step RT-PCR Kit (Qiagen Hilden, Germany) according to the manufacturers’ instructions. Several primer pairs (Table 1) were designed to match the LJV sequence (GenBank MH384278). PCR amplicons ranging from 500 to 800 bp were purified using the Monarch PCR purification Kit (New England Biolabs, Ipswich, MA, USA), and sequenced by Eurofins Genomics (Ebersberg, Germany). A 5′-RACE protocol using terminal deoxydeoxynucleotidyl transferase (New England Biolabs, Ipswich, MA, USA) was adapted for LJV. Briefly, first-strand cDNA was synthesized from total RNA using the LJV genome-specific primer LJV_Part_1 and the Omniscript RT Kit (Qiagen Hilden, Germany). The cDNA was precipitated with ethanol and a polyadenylated or polycytidylate tail was added. Another genome-specific primer (LJV_RACE_1) was then used in conjunction with a primer containing oligo-dT/oligo-dG and adapter sequences (T1/T2) to amplify the LJV 5′-end. Finally, nested PCR was carried out using primers hybridizing to the adapter (T22) and 5′-terminal LJV sequences (LJV_RACE_2). LJV amplicons were then transferred to vector pGEM-T easy (Promega, Madison, WI, USA). The LJV-Ds-OS20 sequences were submitted to GenBank under accession number MW556743.

### 2.8. Phylogenetic Analysis

LJV-Ds-OS20-ORF sequences were aligned with 56 LJV sequences in GenBank using Geneious v10.2.6 (https://www.geneious.com/, accessed on 14 September 2020) and the ClustalW algorithm, and were manually edited to correct possible errors. A phylogenetic tree was then constructed using the neighbor-joining method [26]. Whole genomes were not available from all viral strains and the calculated alignments of the genome fragments did not always fit well. So we decided to use a gene fragment from a more conserved region, where our control outgroup was clearly separated, to show the relationships between the strains. A 483-bp sequence encoding the RNA-dependent RNA polymerase (RdRp) of LJV from drosophilids (*n* = 18) and *Apis mellifera* (*n* = 1) was aligned using the RdRp of deformed wing virus (DWV) strain DWV-USA-p119 (*n* = 1) as the outgroup. The alignments were edited manually where necessary, using the conserved protein domains as a guide. Bootstrap analysis was based on 1000 replicates.

## 3. Results

### 3.1. LJV Production System

Isolate LJV-Ds-OS20 was first identified in Ockstadt, Germany [11] and was selected for analysis because no other virus species were detected in the initial stock. LJV-Ds-OS20 was passaged under laboratory conditions by microinjection into adult Ds flies. The resulting master stock tested negative for the well-known *Drosophila* A virus, *Drosophila* C virus, cricket paralysis virus, Flock House virus and invertebrate iridescent virus 6 (IIV6) based on established RT-PCR protocols [11]. Adults, pupae and larvae were evaluated as laboratory production hosts for LJV by analyzing the extracts from individual insects by qRT-PCR, thus avoiding errors caused by injection failures. The injection of the insects with 4.6 × 10^5^ GE (46 nL) from a master stock resulted in substantial virus growth after 3 days in adult female flies, with 1.2 × 10^9^ GE per insect (95% CI +/− 2.2 × 10^8^) detected by qRT-PCR. In contrast, the injection of larvae produced much lower yields of 2.4 × 10^6^ GE per insect (95% CI +/− 4.5 × 10^5^), and the injection of pupae was even less productive, with yields of 2.0 × 10^6^ GE per insect (95% CI +/− 1.1 × 10^5^, Figure 2A). The injection of Ds adults, pupae, and larvae with the LJV-Ds-OS20 master stock killed all individuals within 3–4 days (Figure 2B) and showed an infection rate of 100%. We decided to use adult Ds flies as hosts for virus propagation because they provide the best production yields.

### 3.2. LJV-Ds-OS20 Sequence Analysis

We harvested virus particles by centrifuging the homogenates of infected adult flies and passing the supernatants through a syringe filter. The consensus sequence of the LJV-Ds-OS20 isolate was then determined by Illumina whole-genome sequencing. The closest match was the LJV strain HUNplus645 (MH384285). We then used adult fly hosts to replicate the virus by limiting dilution for further analysis. Using oligonucleotides hybridizing to the LJV strain HUNplus645, we amplified 15 genome-spanning cDNA fragments by RT-PCR, cloned them, and determined the clonal virus nucleotide sequence by Sanger sequencing. We also adapted the 5′ and 3′ RACE protocols to complete the terminal sequences of LJV-Ds-OS20. This revealed additional nucleotides at both ends, which have not been reported for other drosophilid LJV strains. The 5′- terminal stretch of 10 nucleotides (GAAAAGTAGT) that has not been reported in other drosophilid LJV strains, was recently presented for an LJV strain from *A. mellifera* (MT681680), suggesting it may be also conserved in the drosophilid strains, but was not captured by the sequencing method used for these viruses (Figure 3A). This sequence was also not represented in our NGS data sets but could be confirmed by conventional RT-PCR. Similarly, we identified six additional nucleotides (ATATAT) immediately adjacent to the 3′ polyadenylated tail, unambiguously revealed by sequencing using an oligo-dT primer (Figure 3B). The complete LJV-Ds-OS20 genome sequence is 10,266 nucleotides in length and is relatively AU-rich (29.7% A, 28.8% U, 20.7% G, 20.8% C). The genome encodes a single polyprotein (3047 amino acids) with capsid structural proteins occupying the N-terminal third and nonstructural proteins the remainder. The ORF is flanked by a 5′-UTR of 918 nucleotides and a 3′-UTR of 204 nucleotides, excluding the polyadenylated tail (GenBank MW556743).

Phylogenetic analysis based on the RdRp gene (Figure 4) showed that LJV-Ds-OS20 is closely related to other drosophilid LJV strains including the type strain isolated from an Australian line of *Drosophila melanogaster* (Dm; GenBank MH384278) [14]. The close relationship between the German LJV isolate from Ds and the Australian LJV isolate from Dm is notable because other LJV strains from Europe do not cluster in this group, but instead form their own lineages. The newly-discovered terminal nucleotide sequences of LJV-Ds-OS20 match those of the non-drosophilid LJV strain MT681680, which has only a distant relationship (Figure 4). In contrast, the alignment with the Australian drosophilid LJV strain contained only 168 mismatches, resulting in an amino acid sequence identity >99% after in silico translation. Showing the greatest identify (>98%) over the shared sequence of 10,250 nucleotides (Figure 2). Furthermore, among the eight amino acids that differed between the strains, six were conservative changes.

### 3.3. Purification of LJV-Ds-OS20

A virus suspension prepared from 2000 female flies injected with LJV-Ds-OS20 (Figure 1B) was used to characterize the LJV particles. Only successfully injected flies that remained viable 1 day post-injection were included in the experiments, as indicated by a melanization spot at the injection site (Figure 1C), whereas flies that succumbed to the injection were excluded to ensure high viral loads in the homogenate (Figure 1D). The cleared virus suspension was concentrated by ultracentrifugation on a 25% (*w/v*) sucrose cushion and the resuspended pellet was loaded onto a 1.0–1.55 g/mL CsCl density gradient to separate the particles by buoyancy. Basal illumination revealed fuzzy bands in the upper low-density region, which was RT-PCR negative for LJV, and a sharp band in the high-density region at the bottom of the gradient. The latter was withdrawn and the buoyant density of the material was 1.40 g/mL, as determined using a fine balance (Figure 5A). The material from the band was harvested by diluting the CsCl solution, followed by ultracentrifugation and resuspension of the pellet in 100 µL PBS. The presence of LJV was verified by qRT-PCR, revealing a viral load of 5.2 × 10^10^ GE/mL, subsequently used for TEM and SDS-PAGE. The quality of the purified virus preparation was assessed by TEM, revealing large numbers of icosahedral particles with an apparent diameter of 30–35 nm (Figure 5B,C). The particles displayed the typical morphology of other insect picornaviruses, nodaviruses and tetraviruses [27,28,29]. Fractionation of the preparation by SDS-PAGE revealed the presence of two dominant protein bands with apparent molecular weights of 29 and 33 kDa, corresponding to the anticipated size of *Iflavirus* structural proteins (Figure 5D). We therefore decided to use the purification strategy described above for the further analysis of LJV capsid proteins.

### 3.4. Mapping of the Structural Protein Region

The capsid proteins from purified LJV-Ds-OS20 particles were separated by 2D SDS-PAGE and blotted onto a PVDF membrane for N-terminal sequencing by Edman degradation and peptide mass fingerprinting by MS following tryptic digestion (Figure 6). The N-terminal sequence of VP2 (MEDDGGQGDT) was mapped to amino acid residue 171 of the translated LJV-Ds-OS20 polyprotein. This corresponded to spots V4 and V5, with an apparent molecular mass of 29 kDa (pI = 6.0 and 6.5, respectively). The N-terminal sequence of VP1 (DKPYDDQRVQ) was mapped to amino acid residue 433 of the polyprotein. This corresponded to spots V1, V2, and V3 with apparent molecular masses of 31, 33, and 47 kDa (pI = 5.25–5.9). Protein bands excised from the 1D-gel were also subject to N-terminal sequencing, and band B1 (Figure 5D), with an apparent molecular mass of 33 kDa, also featured the sequence DKPYDDQRVQ. The N-terminus of VP3 could not be determined by Edman degradation because no spots with sufficient amounts of this capsid protein could be identified. The N-terminal sequence of band B2 from the 1D gel also could not be resolved.

MS data obtained from the protein spots confirmed the N-terminal sequencing results (Figure 7). The detection of VP2-specific peptides in spots V4 and V5 resulted in the coverage of five sequence blocks in the molecule spanning from the known N-terminus to the C-terminal region. VP4 peptides were not detected, so the mapping of this hypothetical gene product was based solely on the typical cleavage site and homology to other iflaviruses and picornaviruses. VP1-specific peptides were detected in spots V1, V2 and V3, covering several regions of the protein. Spots V1 (33 kDa) and V3 (31 kDa) led to the identification of peptides from the N-terminal two thirds of the protein, whereas spot V2 (47 kDa) covered the C-terminus of VP1 with a calculated molecular weight of 47.8 kDa (Figure 7). The different apparent molecular weights of the proteins in these spots and the identical N-terminal sequences allowed us to infer the internal cleavage of VP1 (Figure 6). The observed molecular weights and homology fit well with the processing of VP1 in the related *Iflavirus* SBV, but in the absence of further experimental data we were unable to pinpoint the additional C-terminus of VP1. Peptides corresponding to VP3 were found in spot V1, which also contained VP1. Therefore, the precise N-terminal boundaries of the VP3 gene remain unclear, with the detected peptides providing the constraints. Because only a single well-conserved cleavage site was found in this region, a 3C cleavage at the indicated position seems likely. The peptides we detected covered four sequence blocks in the protein, including a C-terminal peptide that could not have been generated by trypsin cleavage, thus representing the end of the structural protein region of the LJV polyprotein.

### 3.5. Protease Cleavage Sites

The N-terminus of VP2 and VP1 and the C-terminus of VP3 were definitively identified. These bona fide targets of the 3C protease within the polyprotein provided insights into its cleavage site specificity. The Q_171_/ME cleavage site at the VP2 N-terminus and the Q_1113_/ME cleavage site at the VP3 C-terminus follow the typical pattern of picornavirus 3C proteases, which cleave immediately downstream of glutamine or glutamic acid residues [30]. The cleavage site residues at the VP1 N-terminus and VP3 C-terminus are highly conserved among all LJV strains (Figure 8). In addition, the Q/M motif together with a proline or alanine at position −4 and an aspartic acid residue at +3, as seen for the VP1 N-terminal and VP3 C-terminal sites, were also present at the postulated VP2/VP4 and VP1/VP3 cleavage sites. A somewhat different motif (VRAQ_1170_/MIAE) is found in the nonstructural protein region that separates the known helicase domains from the polyprotein. No further putative 3C protease cleavage sites with the Q/M motif were found within the non-structural protein region, except Q_1740_/M, suggesting the LJV 3C protease has a broader specificity. The VP1 N-terminus appears to be generated by a different protease, as claimed for other picorna-like viruses, with the target site M_423_/D. This site also resembles a classical pattern with an aspartic acid residue at +1 and a proline residue at +3. The N-terminal VP1 cleavage site is very similar to cleavage sites found in other well-characterized iflaviruses, such as DWV.

## 4. Discussion

We isolated a strain of LJV from Ds specimens collected in Ockstadt, Germany. Strain LJV-Ds-OS20 was cloned by limiting dilution and produced under standardized conditions in a laboratory-bred Ds colony (Figure 1). Particles extracted from the infected flies were concentrated by ultracentrifugation, and total RNA was prepared for Illumina sequencing. We generated a high-coverage consensus sequence and synthesized oligonucleotides suitable for amplification. Using RT-PCR, RACE-PCR and molecular cloning, we then determined the complete genome sequence of strain LJV-Ds-OS20.

The structural proteins of LJV-Ds-OS20 are located in the N-terminal third of the polyprotein behind a leader protein sequence, whereas nonstructural proteins are found in the C-terminal two thirds, as described for other iflaviruses and supported by the presence of multiple conserved domains [23,31]. Phylogenetic analysis of the LJV-Ds-OS20 sequence and other LJV strains showed a close genetic relationship between the drosophilid LJV strains, with an Australian strain (MELplus20518, MH384324) The other near-complete LJV sequences in the database were generated using next-generation sequencing methods that often struggle to complete homopolymer runs, and this may explain why the extra nucleotides are missing in these cases. Comparison with other LJV genomes revealed a general low diversity at the 5′ termini, but greater diversity was seen when comparing other iflaviruses [32,33,34].

*Iflavirus* capsid proteins are named based on the ranking of molecular weights from the largest (VP1) to the smallest (VP4) [19,22,35]. An alternative approach uses the homology of these proteins to their counterparts in classic mammalian picornaviruses, reflecting their functions in morphogenesis [21,23,36]. We used the traditional weight ranking system herein because it facilitates comparisons with previous studies. VP1, VP2 and VP3 assemble into a capsid protomer, which is arranged in a pseudo-T3 icosahedral symmetry to form the *Iflavirus* capsid [23]. For the analysis of LJV-Ds-OS20 structural proteins, we purified the concentrated virions by CsCl gradient centrifugation. Particles of a uniform size (~30 nm) and icosahedral morphology were visible in TEM images, indicating successful purification (Figure 4B,C). Based on the analysis of protein sequences, we constructed a proteolytic cleavage map within the deduced polyprotein sequence (Figure 6). The virus-encoded 3C-like proteinase is predicted to target five dipeptide cleavage sites (Q_171_/M, Q_408_/M, M_432_D, Q_856_M, and Q_1113_M) that define the boundaries of five proteins with the typical gene order adjacent to the non-structural protein region: L (p20.3)–VP2 (p26.3)–VP4 (p2.3)–VP1 (p47.8)–VP3 (28.9) [31,37]. The identification of the VP2 N-terminus allowed us to define the boundaries of the L-protein (amino acids 1–171). Neither the precise C-terminus of VP2 nor the N-terminus of VP4 could be determined experimentally, which can also be explained by the fact that a VP4 does not seem to occur in the virions of the iflaviruses. Boundaries of VP2 (172–408) and a hypothetical VP4-like peptide (409–432) can be deduced from the presence of a peptide close to the C-terminus of VP2 and the typical cleavage site downstream of the peptide. The identification of the VP1 N-terminus, the detection of several molecular weight variants, and the presence of a conserved cleavage site suggest a larger VP1 pro-protein (433–856) is processed into a mature VP1 protein (approximately 433–724) with the precise C-terminus unknown. For VP3, the typical N-terminal cleavage site and the C-terminus identified by peptide mass fingerprinting indicate the sequence range 857–1113. Compared to VP1 and VP2, we detected a much lower quantity of VP3 in the preparation, suggesting issues with isoelectric focusing or separation by SDS-PAGE.

The determination of N-terminal sequences for VP2 and VP1 and the detection of a C-terminal peptide for VP3 define the 3C-like protease cleavage sites within the polyprotein and provide insights into the protease activity associated with polyprotein processing. The LJV 3C-like protease cleaves dipeptides at a Q (–1) to M (+1) cleavage site, which is highly conserved in all known LJV isolates (Figure 7). This cleavage pattern also matches the classic pattern for viral 3C-like proteases, which tend to cleave the peptide bond downstream of glutamine (Q) or glutamic acid (E) residues. The flanking residues are usually conserved for individual 3C proteases and are similar to those found in other iflaviruses, with a preference for a negatively charged residues at position +2 (E or D) and a proline (P) at position –4. As reported for most known picorna-like viruses, the maturation of LJV VP1 shows a divergent cleavage pattern at the site –4 IKNM/DKPY +4. This suggests another protease is responsible for cleavage at this site, or the 3C protease has an atypical additional specificity. Comparison with the VP1 proteins of other iflaviruses revealed that the VP1 N-terminus of LJV is typical: the aspartic acid (D) residue at position +1 and the proline (P) residue at position +3 are highly conserved among all known iflaviruses. We observed considerable heterogeneity among the LJV VP1 proteins, suggesting that additional processing steps in the host involve autocatalysis or specific host protein-mediated proteolysis to remove the C-terminus (Figure 6). The VP2 spots with different pI values may represent post-translational modification variants, such as phosphorylated and non-phosphorylated versions. However, we cannot exclude the possibility that the heterogeneity reflects artifacts of the purification process.

The direct comparison of our LJV polyprotein map and other iflaviruses revealed some similarities (Figure 9). First, the preferred cleavage sites of the LJV 3C-like protease (Q/ME) are similar to the Q/MD sites preferred by SBV. Second, the molecular weight of the LJV capsid proteins is similar to the homologous proteins from DWV and SBPV. However, the unusual C-terminal part of LJV VP1 is represented by a so-called protrusion domain in DWV and SBPV, forming a globular structure on the capsid surface, whereas in SBV [21,38] the C-terminus of VP1 is processed further to form an independent minor capsid protein (MICP) attached to the capsid surface [23]. We detected C-terminal VP1 peptides in the 47.8 kDa pre-protein, but not in the mature 33 kDa form. Based on the apparent molecular weights and sequences, we speculate that LJV VP1 is processed at position S724/H similar to the SBV cleavage site. We found no evidence of a 15 kDa MICP in our LJV preparations, but if such a protein were only loosely bound to the capsid surface, it may have been lost during purification. It would therefore be interesting to use X-ray diffraction or the gentler method of cryo-electron microscopy to draw definitive conclusions about the position of the VP1 C-terminus in LJV.

To conclude, we have characterized the structure of the LJV-Ds-OS20 strain that severely reduces the lifespan of Ds following infection, making it a suitable candidate biopesticide [11]. This could help to reduce the economic burden of Ds in the agricultural industry, where losses to fruit crops are valued at millions of euros every year [39,40,41]. Traditional control methods are hindered by the short life cycle and high reproductive rate of Ds, the rapid acquisition of pesticide resistance and the protection of eggs and larvae within the ripening fruit. Most iflaviruses appear to cause asymptomatic infections in insects, although previous studies have demonstrated the virulence of iflaviruses that infect honeybees and silkworms [18,42]. To facilitate the design of biopesticides based on LJV, we characterized the major structural proteins of the virion and their relationship to the polyprotein, which could facilitate the development of application-specific particles with higher or lower stability in the environment using reverse genetics or chemical modifications [43].

## Figures and Tables

**Figure 1 viruses-13-00740-f001:**
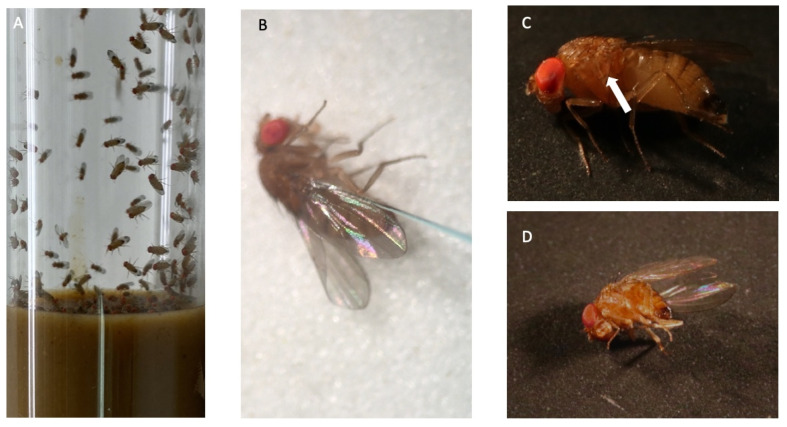
Production system for La Jolla virus strain LJV-Ds-OS20. (**A**) Culture of *D. suzukii* in a 2.5 cm diameter plastic tube containing soybean and cornmeal medium. (**B**) Microinjection with LJV-Ds-OS20 results in (**C**) melanization at the injection site (arrow) the day after the procedure and then (**D**) death of the infected fly 3 days later.

**Figure 2 viruses-13-00740-f002:**
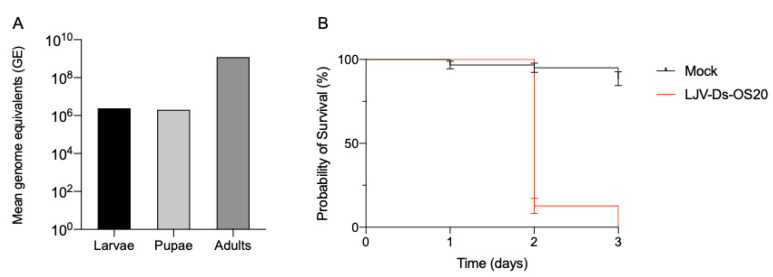
LJV-DsOS20 virus production system. (**A**) LJV was injected in larvae, pupae and adult flies, showing higher virus genome equivalents in adults than in pupae and larvae. (**B**) When virus was injected in female flies, fast mortality was observed in infected flies. Experiments have been performed three times, *p* < 0.001 for log-rank test.

**Figure 3 viruses-13-00740-f003:**
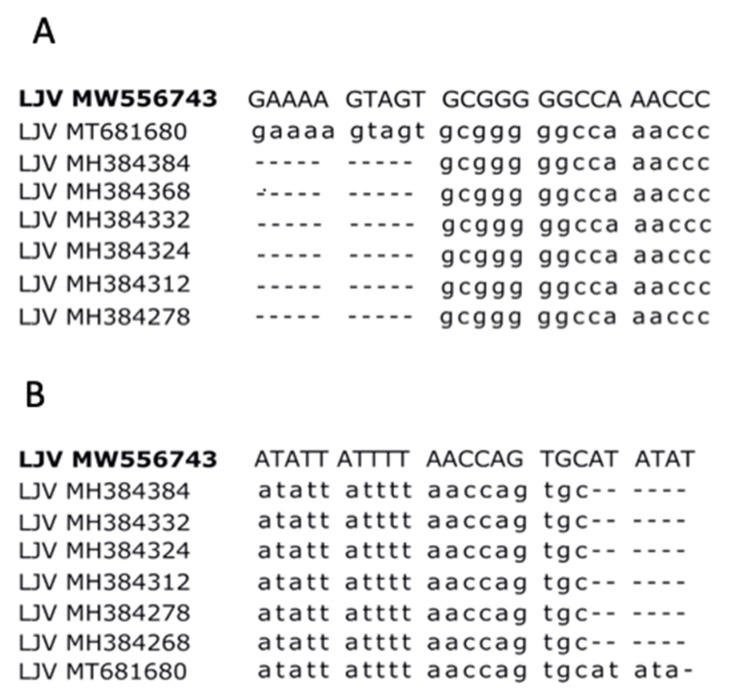
Comparison of the 5′-UTR and 3′-UTR sequences of known La Jolla virus (LJV) strains. (**A**) The 5′-UTR of LJV-Ds-OS20 (LJV MW556743) compared to seven other LJV strains. (**B**) The 3′-UTR of LJV-Ds-OS20 (LJV MW556743) compared to seven other LJV strains.

**Figure 4 viruses-13-00740-f004:**
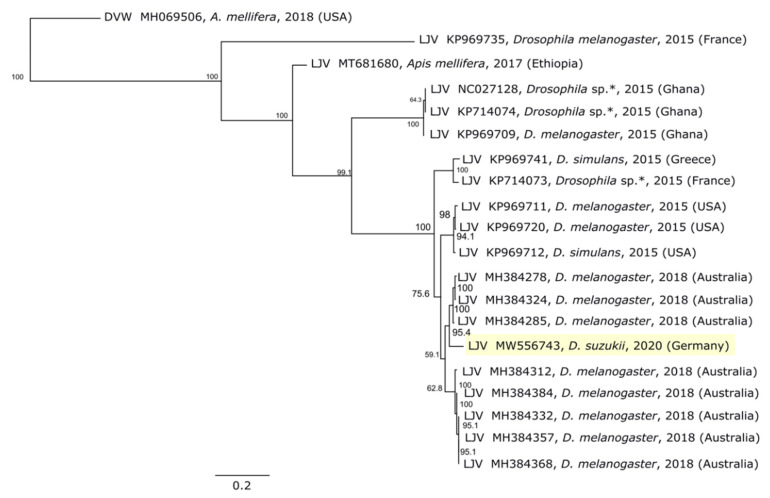
Phylogenetic analysis of La Jolla virus (LJV) strains. The phylogenetic tree was constructed using the neighbor-joining method based on a ClustalW alignment of the 483-bp RdRp gene from published full-length drosophilid LJV sequences (*n* = 18) and from Apis mellifera (*n* = 1). DWV was used as an outgroup. An asterisk (*) indicates sequences obtained from pooled Drosophila. The bar indicates the number of substitutions per site. Bootstrap values on each node reflect 1000 replicates.

**Figure 5 viruses-13-00740-f005:**
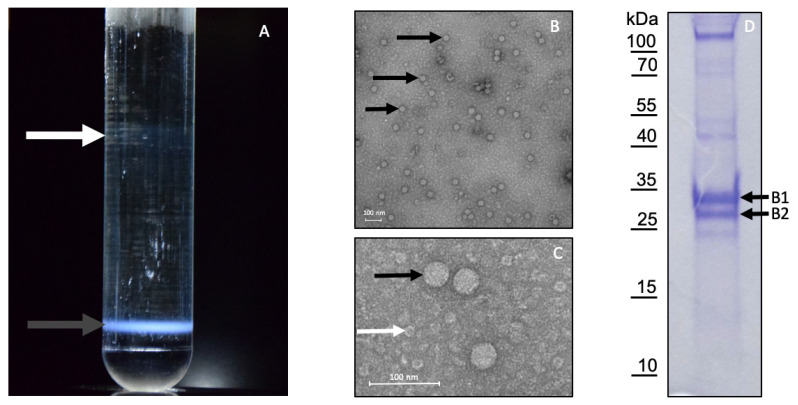
Purification and analysis of La Jolla virus (LJV) particles. (**A**) CsCl gradient purification. The white arrow indicates turbid band of RT-PCR negative material with a buoyant density of 1.15–1.20 g/mL. The black arrow indicates a sharp band of LJV virions with a buoyant density of 1.36 g/mL. (**B**) Transmission electron micrograph of purified LJV particles in negative contrast at a magnification of 85,000×, with black arrows indicating icosahedral structures ~30 nm in diameter. (**C**) Transmission electron micrograph of purified LJV particles in negative contrast at a magnification of 140,000×, with the black arrow indicating icosahedral structures ~30 nm in diameter and the white arrow indicating co-purified material, possibly virion fragments. (**D**) SDS-PAGE analysis of purified LJV particles stained with Coomassie Brilliant Blue. Two dominant protein bands are visible (B1 and B2). Molecular mass markers are shown on the left.

**Figure 6 viruses-13-00740-f006:**
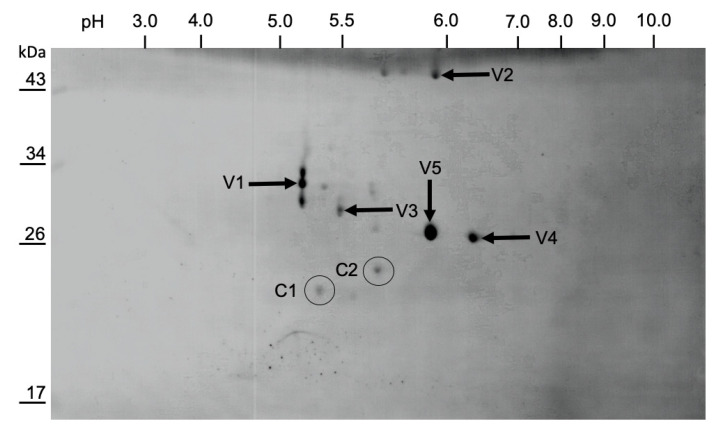
Two-dimensional gel electrophoresis of purified La Jolla virus (LJV) virions. Isoelectric focusing in the pH range 3.0 to 10.0 was followed by SDS-PAGE in a 12.5% polyacrylamide gel and blotting onto a PVDV membrane. The stained protein spots (shown here in grayscale for contrast enhancement) were picked from the membrane for Edmann degradation and peptide mass fingerprinting. The spots labeled V1–V5 are viral proteins and those labeled C1 and C2 are contaminating cellular proteins. Spots V1, V2 and V3 were assigned to VP1 (spot V1 also contains VP3). V4 and V5 were assigned to VP2. The horizontal axis of the membrane indicates pH and the vertical axis indicates molecular size markers.

**Figure 7 viruses-13-00740-f007:**
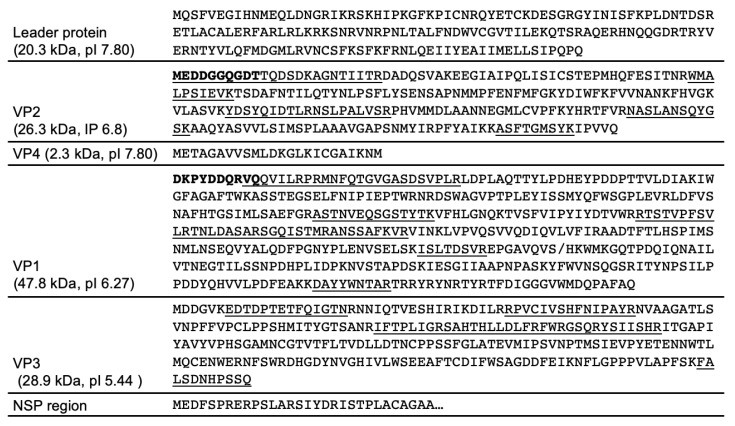
Processing of the La Jolla virus (LJV) polyprotein in the structural protein region of strain LJV-Ds-OS20. The C-terminal cleavage site of the leader protein was deduced from the N-terminus of VP2. The N-terminus of VP2 was determined both by N-terminal sequencing (NTS) and peptide mass fingerprinting (PMF). The precise C-terminus of VP2 was not resolved by PMF. Peptides of a hypothetical VP4 were not detected in our LJV particles. The N-terminus of a VP4 related peptide seems to be reliable determined based on the detectable VP2 peptide located just upstream of the supposed N-terminus and due to a conserved Q/ME cleavage site. The unusual C-terminal cleavage site between VP4/VP1 is deduced by the N-terminus of VP1, which was determined by NTS but not PMF. Neither the exact C-terminus of VP1 nor the N-terminus of VP3 was experimentally mapped. However, by detecting neighboring peptides and a typical Q/MD cleavage site, we can make a prediction. The detection of a VP3 C-terminal peptide lacking basic amino acid residues and the typical cleavage site allowed the precise mapping of the VP3 C-terminus. Amino acids shown in bold were determined by NTS and underlined sequences are peptides detected by PMF. Protein names are provided together with the calculated molecular weights and isoelectric points (pI).

**Figure 8 viruses-13-00740-f008:**
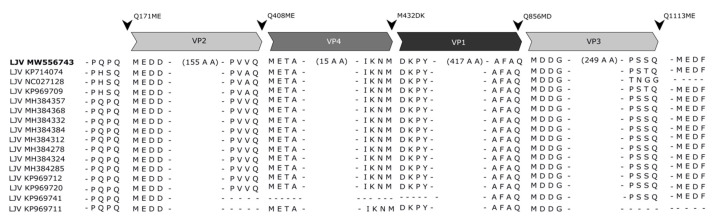
Organization of the structural protein region in the polyproteins of different strains of La Jolla virus (LJV). The capsid proteins (VP2, VP4, VP1 and VP3) are indicated in grayscale. Arrowheads mark cleavage sites as annotated.

**Figure 9 viruses-13-00740-f009:**
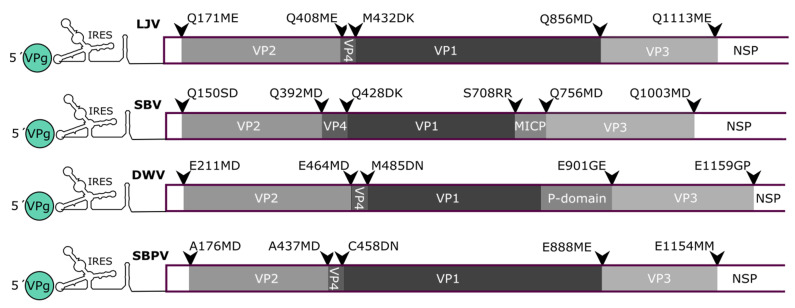
Organization of the structural protein region of different *Iflavirus* polyproteins. We compared the sequences of La Jolla virus (LJV) to sacbrood virus (SBV), deformed wing virus (DWV) and slow bee paralysis virus (SBPV). The map shows the 5′-UTR (including the IRES element and the VPg protein as a green circle), the leader protein (white bar) and the capsid proteins VP2, VP4, VP1 and VP3 in grayscale. The unusual minor capsid protein (MICP) from SBV, corresponding to the processed protrusion (P) domain present in DWV, is also shown. The start of the non-structural protein region (NSP) is indicated in white. Arrowheads mark cleavage sites, which are similar between LJV and SBV.

**Table 1 viruses-13-00740-t001:** Oligonucleotides used in this study.

Oligo Name	FW/RV	Sequence (5′→3′)	Position
LJV_Part_1	FW	GCTCTGAAGGCCTTGGAAAC	22–41
LJV_Part_1	RV	ACGTATCGGGTTCTGTCTCC	721–702
LJV_Part_2	FW	GTGTGTGGCGTTACGATTCT	639–658
LJV_Part_2	RV	TGCGAGGTCCATCATAACATG	1334–1314
LJV_Part_3	FW	TCATACCAAATCGATACACTCCG	1260–1282
LJV_Part_3	RV	ATGTACTCAAGAGGCGTCGG	1960–1941
LJV_Part_4	FW	GAGGAACCGTGATAGCTGGG	1913–1932
LJV_Part_4	RV	ATCTTCGAGTCAGGAGCAGT	2660–2641
LJV_Part_5	FW	CCTGGAGCTGTGCAAGTTTC	2500–2519
LJV_Part_5	RV	AAAGTAACCGTGCCGCAATT	3341–3322
LJV_Part_6	FW	TTCCGATTTTGGCGTGGTTC	3223–3242
LJV_Part_6	RV	CGCCCCATGTTTGTAAGCAA	4019–4000
LJV_Part_7	FW	TCAACAGGCCTTAACATCACT	3940–3860
LJV_Part_7	RV	CTGGCGAACACCTTAAGTCG	4696–4677
LJV_Part_8	FW	CGGCGTGTTATTGATCTTCCA	4585–4560
LJV_Part_8	RV	TTCGACGTTCTTGGTTGTCA	5323–5304
LJV_Part_9	FW	TGAGTGACGAAGAGAGTTTGTC	5168–5189
LJV_Part_9	RV	TGGGACATTACAAGGACGGG	5870–5851
LJV_Part_10	FW	CCAATCAGTAGGTCCGTGGT	5802–5821
LJV_Part_10	RV	TCCGAGTCATTCTGTGCTGT	6527–6508
LJV_Part_11	FW	GCACGTATTCCACCTACAGC	6493–6512
LJV_Part_11	RV	TCAGGTGACGCTCATTTCCT	7303–7284
LJV_Part_12	FW	TCCCGGCATCTCAAAGTGAA	7199–7218
LJV_Part_12	RV	AGGTAAGTGCATTTTGGCCG	7947–7947
LJV_Part_13	FW	TGAAGTTTCCGGTAAGTGCG	7869–7888
LJV_Part_13	RV	AAAGGAGATGCGCAGAACAC	8576–8557
LJV_Part_14	FW	TCCTGAAGTTGCGAACCAGA	8421–8440
LJV_Part_14	RV	TCGCCGTAAACATACATGCA	9134–9115
LJV_Part_15	FW	GGTGAAGGAAAATGGCGTGA	9079–9098
LJV_Part_15	RV	TGGATGTGGCACGAAATTACA	9312–9292
LJV_RACE_1	RV	GGATTCCAAGAGGTAGTCCCGTGAAC	241–216
LJV_RACE_2	RV	CTTTTAGGTGTGGTAGAGTATCATG	153–129
T1		GGCCACGCGTCGACTAGTACTTTTTTTTTTTTTTTTT	Oligo(dT)
T2		GGCCACGCGTCGACTAGTACGGGGGGGGGGGGGG	Adapter
T22		GGCCACGCGTCGACTAGTAC	Adapter

## Data Availability

The data presented in this study are available on request from the corresponding author.

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
