# Peer review of "Organization of the Structural Protein Region of La Jolla Virus Isolated from the Invasive Pest Insect Drosophila suzukii"

_viruses, 2021, doi:10.3390/v13050740_

Round 1
Reviewer 1 Report
Summary: In this study, Carrau et al. seek to study La Jolla virus towards developing biopesticides for Drosophila suzukii (Ds). Ds is a major crop pest and ways to control it are needed to ensure food security. The study is well done and their conclusions are supported by their data. The paper is well written overall. The discussion could be improved by focusing more on what the data mean and their implications.
Major Points:
- The discussion repeats the results and doesn’t always interpret these data. I would suggest focusing more on what the implications of your data are and how they fit within the existing literature.
Minor Points:
- In the introduction you say, “but the genomic organization of its capsid protein sequences is unclear”. What do you mean by this statement? Surely it is well known where in the genome the particular capsid gene lie?
- Line 87 – It’s unclear how the photoperiod and Fig. 1A are connected? Maybe change the position referencing Fig. 1A?
- Line 168 – It says Illumina sequencing was carried out. Can you describe the library prep process and the sequencing? How did you analyze the data? Did you submit the amplicons to Eurofins for library prep + sequencing? Can you please clarify this in the manuscript? Data analysis methods should also be added.
- Line 192 – Why only use this 483-bp fragment? Wouldn’t full genome be more informative?
- Line 203 – Looks like there is an extra space but could be a weird formatting thing.
- Line 205 – Fig. 1C does not correspond to death. Also, do you have survival curves you can add? This would be a lot more informative than only having pictures of the flies. The pictures are awesome so I’m not suggesting you remove them. It would be nice to have some data to go along with them.
- Line 210 – Can you please include the 95% CI for these values. It will be useful to know for others who are performing this type of work what to expect. Also, what was the infection rate for each group? Do you have virus growth over time? These would all be really useful to have.
- 8 – It’s nice to have the accession numbers but it would also be useful to know the virus names. For example. Which one is DWV?
- Line 360 – Maybe collected instead of “captured”? It sounds a bit strange.
- Line 383 – Were you able to go back and map the Illumina reads to the reference sequence obtained after RACE? It would be nice to know if you have additional support from that sequencing data for the presence of the, GAAAAGTAGT. How do you know, for example, that your sequence is correct and the others groups’ is not? It would be nice to have this confirmation for both 5’ and 3’ end.
- “We did not test for the presence of a genome-linked protein (VPg), a heterogeneous class of small proteins bound covalently to the 5′ termini of many RNA virus genomes [35–37].” – This seems out of place. How is this related to the following sentence about Sanger sequencing? Is it necessary?
- “We prefer the homology-based nomenclature” – This seems unnecessary since you don’t use this system.
Reviewer 2 Report
This manuscript describes, to my knowledge, the first laboratory propagation of La Jolla virus in live organisms. Material purified from such infections was used to characterize the virion further, and although there is ample room for further study and repetition, the present report describes significant data of interest to those interested in iflavirus biology.
In a perfect world the authors would have repeated their 1D PAGE mass spec until it works, would identify the faint bands as well, would show 2D class-averages for negatively-stained particles, and would checked their inoculum for infectivity in melanogaster and/or S2 cells. However, my opinion is that the presented experiments represent a worthwhile advance over the current state of the art and provide sufficient backing to the main conclusions of the manuscript.
Minor comments:
polyadenylate -> polyadenylated
What is the concentration of material used for TEM & for PAGE?
Should be mentioned that FastPrep is a “bead beater”-type homogenizer.
What’s the inoculum titer for ll. 204-205, “The infection of Ds adults, pupae, and larvae 204 with a LJV-Ds-OS20 inoculum killed all individuals within 3–4 days” ?
How do you know the other band in Fig 5A is cellular material and not top component? Has some analytical experiment been performed on it?
Fig 4 columns are not aligned. Use of color may make this more intuitive
is VP4 even present in this virion? Can you confirm VP4 exists more than transiently and is structural? If not, it should be explicitly tagged as a hypothetical protein in this case.
What are the identities of C1 and C2? Are they consistently recovered?
Phylogenetic results and comparison to MELplus20518 belongs in Results section.
Although I do not wish to discourage the authors from their implied pursuit of atomic-resolution characterization, in all likelihood this would not reveal the VP1 C-terminus because, for several other nonenveloped viruses exhibiting a mosaic composition of isoforms or cleaved forms, asymmetric reconstruction has not been sufficient to reveal the heterogeneous or flexible termini.
